# Pre-collecting lymphatic vessels form detours following obstruction of lymphatic flow and function as collecting lymphatic vessels

**Kimi Asano**[1,2☯‡], **Yukari Nakajima**[3☯‡], **Kanae Mukai**[3], **Tamae Urai**[4], **Mayumi Okuwa**[3], **Junko Sugama**[5], **Chizuko Konya**[6], **Toshio Nakatani**[3]*

**1** Department of Clinical Nursing, Graduate Course of Nursing Science, Division of Health Sciences, Graduate School of Medical Sciences, Kanazawa University, Kanazawa, Japan, **2** School of Nursing, Kanazawa Medical University, Uchinada, Japan, **3** Faculty of Health Sciences, Institute of Medical, Pharmaceutical and Health Sciences, Kanazawa University, Kanazawa, Japan, **4** Faculty of Nursing, Toyama Prefectural University, Toyama, Japan, **5** Advanced Health Care Science Research Unit, Innovative Integrated Bio-Research Core, Institute for Frontier Science Initiative, Kanazawa University, Kanazawa, Japan, **6** Faculty of Nursing, Ishikawa Prefectural Nursing University, Kahoku, Japan

☯ These authors contributed equally to this work.
‡ These authors are co-first authors on this work.
* nakatosi@staff.kanazawa-u.ac.jp

**Data Availability Statement:** All relevant data are within the paper.

## Abstract

### Background

Previously, we showed that lymphatic vessels (LVs) formed detours after lymphatic obstruction, contributing to preventing lymphedema. In this study, we developed detours using lymphatic ligation in mice and we identified the detours histologically.

### Methods and results

Under anesthesia, both hindlimbs in mice were subcutaneously injected with Evans blue dye to detect LVs. We tied the right collecting LV on the abdomen that passes through the inguinal lymph node (LN) at two points. The right and left sides comprised the operation and sham operation sides, respectively. Lymphography was performed to investigate the lymph flow after lymphatic ligation until day 30, using a near-infrared fluorescence imaging system. Anti-podoplanin antibody and 5-ethynyl-2'-deoxyuridine (EdU) were used to detect LVs and lymphangiogenesis. Within 30 days, detours had developed in 62.5% of the mice. Detours observed between two ligation sites were enlarged and irregular in shape. Podoplanin⁺ LVs, which were located in the subcutaneous tissue of the upper panniculus carnosus muscle, connected to collecting LVs at the upper portion from the cranial ligation site and at the lower portion from the caudal ligation site. EdU⁺ cells were not observed in these detours. The sham operation side showed normal lymph flow and did not show enlarged pre-collecting LVs until day 30.

### Conclusions

Detours after lymphatic ligation were formed not by lymphangiogenesis but through an enlargement of pre-collecting LVs that functioned as collecting LVs after lymphatic ligation.

**Funding:** This work was funded by JSPS KAKENHI Grant Number 18H03072 to T N and 16K11931 to KA, https://www.jsps.go.jp/; and The Yasuda Medical Foundation to YN, http://www.yasuda-mf.or.jp/. The funders had no role in study design, data collection and analysis, decision to publish, or preparation of the manuscript.

**Competing interests:** The authors have declared that no competing interests exist.

Further studies are required to explore the developmental mechanism of the lymphatic detour for treatment and effective care of lymphedema in humans.

## Introduction

Insufficiency of the lymphatic transport system due to abnormal lymphatic vessel (LV) or damaged lymphatic system causes stagnation in protein transportation and interstitial fluid flow, resulting in lymphedema [1]. Breast cancer-related lymphedema has been reported as the most common form of lymphedema, globally [2]. As lymphedema develops, patient's quality of life is reduced. Complications such as bacterial infection and skin pathologies can occur in patients with lymphedema, and they are required to undergo arduous self-care in daily life. Radical treatment, prevention options, and effective care strategies have not yet been established; therefore, exploring treatment options in clinical settings is an important goal.

Lymphoscintigraphy is the primary imaging modality used in determining the diagnosis of suspected extremity lymphedema in patients. Lymphoscintigraphy, magnetic resonance imaging (MRI), and lymphangiography findings in patients with lymphedema show interruption of lymphatic flow, collateral lymph vessels, dermal back flow, delayed flow, delayed visualization or non-visualization of lymph nodes (LNs), dilated lymphatics, and no visualization of the lymphatic system [3–9]. When collateral LVs bridging the distal LV (over the area of lymphadenectomy) to the proximal LV or to the remaining LNs with LVs are observed in patients using lymphoscintigraphy (at the area of the lymphadenectomy), lymphedema is said to be mild or moderate [3, 5, 7, 10–12]. Moreover, animal studies involving canines, rabbits, rats, and mice have previously reported that similar collateral LVs (also known as lymphatic detours) develop following LN dissection alone or in combination with LVs [13–22]. Collateral LVs in rodents have been detected using indocyanine green (ICG) in 2 days [22], in 3 days [20], in 1 week [14, 15] and in 10 days [21] after lymphadenectomy. In our previous study, mice showed well-developed detours in the abdomen after lymphadenectomy of the inguinal LN [21]. We concluded that these detours prevent lymphedema in mice because these mice did not develop lymphedema. Therefore, it is important to elucidate the structure, development, and function of collateral LVs or lymphatic detours.

The lymphatic system in the skin consists of initial or capillary LVs in the dermis, connecting or pre-collecting LVs in subcutaneous tissues, and collecting LVs in the subcutaneous area of the epimysium. Moreover, deep collecting LVs run alongside deep blood vessels among the muscles. Lymph drainage occurs in this turn (capillary, pre-collecting, collecting, and deep collecting LVs). Therefore, if LVs or LNs are dissected, new LVs are formed by the sprouting of lymphatic endothelial cells from the end of the dissected LVs, i.e. lymphangiogenesis [23, 24] or lymphatic pathway changes, for instance, lymph backflow to the pre-collecting LV from the collecting LV, or the opening of a channel between collecting LVs, occur [25]. Moreover, capillary LVs have been reported to appear in granulation tissue via the sprouting of lymphatic endothelial cells on the remaining LVs approximately 7 days after the full-thickness skin wound was created [26]. Tammela et al. [27] showed that it took 2 months to regenerate the collecting LVs after applying vascular endothelial growth factor (VEGF)-C to a mouse following the removal of the axillary LNs and all of the associated collecting LVs. This indicated that it took a longer time for the new collecting LVs to be formed compared to new capillary LVs. Thus, it is unclear whether the lymphatic detours are newly generated LVs, formed through

lymphangiogenesis or through the changing lymphatic patterns between the remaining or the pre-existing LVs.

Therefore, in this study, we aimed to determine whether lymphatic detours appeared following lymphatic ligation only (where lymphatic detours ran in the skin), without the LNs, using ICG and by observing histological serial sections. We also determined whether the lymphatic detours were pre-existing LVs or newly generated LVs histologically by immunohistochemical (IHC) staining. We consider that the knowledge gained concerning the collateral LVs in patients with lymphedema could help to reduce lymphedema formation and progression.

## Materials and methods

All experimental protocols and animal care procedures were approved by the Committee on Animal Experimentation of Kanazawa University (AP-153596) and the experimental methods were performed in accordance with its guidelines.

### Animals

In total, 51 BALB/cCrSlc male mice (Sankyo Lab Service Corporation, Inc., Toyama, Japan) were used. The mice were caged individually in an air-conditioned room at a temperature of 25.0 ± 2.0 ˚C, and the lights were kept on from 0815 h to 2015 h. Water and pelleted food were given freely. Fifteen mice underwent preliminary experiments while developing the protocols for this study to determine how to ligate the LVs using surgical sutures and to determine the appearance of the lymphatic detours. Based on these preliminary experiments, 8 mice were continuously observed for 30 days for the appearance of lymphatic detours, following lymphatic ligation, through ICG injecting on days 15, 21, 27, and 30. After this observation, all the eight mice were euthanized, and skin, including the lymphatic tissues, were histologically evaluated. Subsequently, 28 mice were used for histological evaluations on days 10, 12, 15, 18, 21, 24, and 27. Each day, 4 mice were euthanized after having confirmed the appearance of lymphatic detours using injected ICG. These 28 mice were not continuously observed using ICG, since the main aim was to obtain histological specimens. Mice were used on days 10 and 15 prior to the appearance of lymphatic detours, and on days 18 and 24, to obtain detailed information from their histological specimens.

### Surgical procedure for lymphatic ligation

In a previous study, we developed detours through lymphadenectomy [21]. However, the experimental method used in this study differed from that of our previous study in that we did not excise the inguinal LN but ligated the collecting LV at the abdomen. Therefore, the appearance pattern of detours using this surgical technique was unexpected; hence, we undertook to perform a preliminary experiment using 15 mice. In the preliminary experiment, we ligated the collecting LV that passes through the inguinal LN (see (a) in Fig 1E) at two points, namely, the cranial ligation site and the caudal ligation site. The caudal ligation site was located approximately 5 mm superior to the inguinal LN. There was a 5 mm distance between the two ligation sites. We found that, after the lymphatic ligation, most detours had appeared between the two collecting LVs in the abdomen (see (a) and (b) in Fig 1E) by day 30. Therefore, we adopted lymphatic ligation to observe the reconstruction of the lymphatic route in the present study.

The mice were anesthetized with 1.75% isoflurane inhalation (FUJIFILM Wako Pure Chemical Corporation, Ltd., Osaka, Japan). Twenty-five μl of 3% Evans blue dye (w/v) was subcutaneously injected into both hindlimbs to detect the LVs in the abdomen that pass through the inguinal LN. Thirty minutes after the dye injection, a 1 cm dorsal incision was

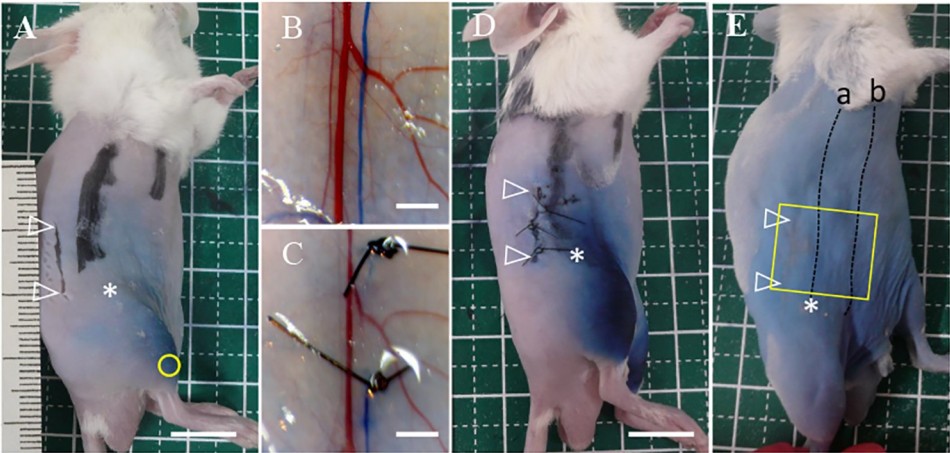

**Fig 1. Surgical procedure for lymphatic ligation and the sectioning area for histological observation. (A)** Evans blue dye was injected into the hindlimb (yellow circle). A 1 cm incision was made 5 mm from the dorsal site in relation to the collecting LVs passing through the inguinal lymph node (LN) on the abdomen at 5 mm from the inguinal LN; scale bar of 1 cm. **(B)** The incised skin was peeled from the epimysium and the collecting LV (stained in blue color) passing through the inguinal LN was exposed; scale bar of 2 mm. **(C)** It was carefully tied at 2 points with 6–0 nylon sutures under a stereoscopic microscope; scale bar of 2 mm. **(D)** Skin edges were sutured with 6–0 nylon sutures at 3 points; scale bar of 1 cm. **(E)** The area, including the incision site, in which the collecting LV (a) passed through the inguinal LN and ran into the axillary LN, and; the collecting LV (b) emerging from the lower abdomen, was harvested for histological observation (yellow square); scale bar of 1 cm. Open white arrowheads: incision sites; asterisk: inguinal LN.

made on the right side (the operation side) at a 5 mm distance from the LVs passing through the inguinal LN (Fig 1A). The skin was peeled from the epimysium of the latissimus dorsi and external oblique muscles to expose the collecting LVs in the subcutaneous tissue (Fig 1B), after which the stained collecting LVs on the right abdomen were carefully tied at 2 points 5 mm apart with a 6–0 nylon suture, under a stereoscopic microscope (Fig 1C). The blue color of the LVs disappeared after the ligation. This ligation was performed without injury to the blood vessels along with the collecting LVs. Skin moisture was maintained by applying saline (0.9%) during lymphatic ligation. Skin edges were sutured with 6–0 nylon sutures at 3 points (Fig 1C). A 1 cm dorsal incision and skin peeling from the epimysium were also performed on the left side (as the control) and the incision was sutured using 6–0 nylon sutures as part of the sham operation. This surgical procedure took approximately 20 minutes per mouse and was performed by a single operator (YN).

## Observation of ICG lymphography

The observation protocol for lymphography has previously been described [20]. In the mice that were anesthetized using 1.75% isoflurane inhalation, we injected 2 μl ICG (2.5 mg/ml) into the hindlimb to investigate the lymph flow after lymphatic ligation. Surgical sites were then observed using an infrared camera system (PDE-neo, Hamamatsu Photonics, Shizuoka, Japan), and fluorescent images were obtained. Examinations of ICG lymphography were conducted on days 10, 12, 15, 18, 21, 24, 27, and 30 after lymphatic ligation.

## Evaluation for lymphangiogenesis after lymphatic ligation

Lymphangiogenesis was evaluated using 5-ethynyl-2'-deoxyuridine (EdU), which is an alkyne-conjugated nucleoside analog of thymidine. EdU is used as an alternative to radioactive thymidine or bromodeoxyuridine (BrdU). In contrast to a BrdU-based unscheduled DNA synthesis

assay, EdU can be directly conjugated to fluorescent azide, and the assay requires neither DNA denaturation nor antibodies [28]. After anesthesia using 1.75% isoflurane inhalation, mice were intraperitoneally injected with 100 μg of EdU twice, that is, 24 h prior to tissue sampling and 3 h prior to tissue sampling. Additionally, the small intestine was harvested to determine whether EdU had been incorporated into the nucleus.

### Tissue processing

After a 25 μl injection of 3% Evans blue dye (w/v) in both hindlimbs, the mice were intraperi-toneally euthanized using a large dose of sodium pentobarbital (0.5 mg/g weight) on days 10, 12, 15, 18, 21, 24, 27, and 30 after lymphatic ligation. The operative field, or the sham operative field, and the surrounding intact skin including two collecting LVs in the abdomen, were harvested (Fig 1E), stapled onto transparent plastic sheets to prevent excessive contraction of the specimens, and fixed with a zinc fixative solution (BD Pharmingen, CA) overnight. Using tweezers, a spacer was placed between the specimens and the sheets to allow the fixative solution to flow. Specimens were incubated overnight successively in phosphate buffered saline (PBS) with 10% sucrose, in PBS with 20% sucrose, and in PBS with 30% sucrose prior to being embedded in OCT compound (Sakura® Finetek, Tokyo, Japan) with distilled water and Tween 20. Frozen tissue was stored at −80 ˚C until further use. Serial sections, 7 μm thick on day 30, were mounted for the detection of detours. Specimens on days 10 to 27 were sectioned at 6 points every 1 mm. Twenty serial sections at each point were subjected to hematoxylin and eosin (H&E) or immunostaining.

### Immunohistochemical staining

Anti-Podplanin antibody was used for detecting LVs because podoplanin is expressed in the capillary as well as in the pre-collecting, and collecting LVs [29, 30]. Specimens for IHC stain-ing of LVs were washed with PBS comprising 0.03% Tween-20 (PBST), incubated with protein blocking solution (X0909; Dako, CA), endogenous avidin/biotin blocking kit (ab64212; Abcam, Cambridge, UK), and 0.03% hydrogen peroxide in methanol. The Syrian Hamster anti-podoplanin/gp36 antibody (ab11936: Abcam, Cambridge, UK) was used at a dilution of 1:600. Subsequently, the specimens were incubated with secondary antibodies linked with bio-tin (ab7145: Abcam, Cambridge, UK) followed by streptavidin linked with HRP (ab7403: Abcam, Cambridge, UK). The EdU staining for evaluation of lymphangiogenesis was per-formed using Click-iT™ EdU Colorimetric IHC Detection Kit (C10644: Invitrogen, OR). Nega-tive control sections were obtained through omitting each primary antibody.

## Results

### ICG lymphography

The injected ICG in the hindlimb drained in a distal to proximal direction. Lymphatic routes after ligation were divided into either detours, original routes, or no detected routes (Fig 2). On day 30, 5 of 8 mice showed detours, 3 of 8 mice showed original routes, and no mice showed no detected routes (Table 1). The number of mice with no detected routes gradually decreased until day 30. In contrast, the number of mice with detours or original routes gradu-ally increased until day 30. During the observation period, the sham operation side showed normal lymph flow passing through the inguinal LN and flowing into the axillary LN (Fig 2A). The detours that appeared on days 15, 21, and 27 remained until day 30. Original routes appeared on days 15, 21, and 27 remained until day 30. The LVs with no detected routes had changed to detours or original routes by days 21, 27, and 30 (Table 1). Well-developed detours

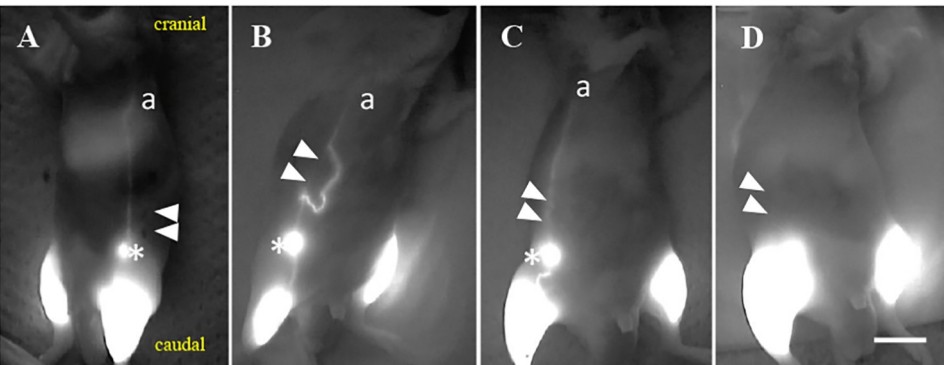

**Fig 2. Representative images of lymphography after lymphatic ligation. (A)** Lymph flow obtained on the sham operation side on day 30. **(B)** Some detours were detected in the abdomen on day 30. These detours originated from the lower portion from the caudal ligation site (closed white arrowhead) of the collecting LV (a) and passed through the inguinal LN and connected to the upper portion from the cranial ligation (closed white arrowhead) site to collecting LV (a) passing through the inguinal LN. **(C)** An original route was detected after lymphatic ligation on day 30. **(D)** No lymph flow was detected on the operation side on day 18. a: the collecting LV that passed through the iliac LN and flowed into the axillary LN; scale bar of 1 cm; asterisk: inguinal LN; closed white arrowheads: ligation sites.

occurred in the lower portion from the caudal ligation site of the collecting LV passing through the inguinal LN and connected to the upper portion from the cranial ligation site of the collecting LV passing through the inguinal LN (Fig 2B). Original routes were also detected in healthy mice after injection of dye into the hindlimb. However, the lymphatic routes between the lower portion from the caudal ligation site and the upper portion from the cranial ligation site seemed to be distorted, as indicated by the white arrowheads in Fig 2C. In mice with no detected routes, the inguinal LN was not shown after an ICG injection into the hindlimb (Fig 2D), whereas mice with detours or original routes showed the inguinal LN. The three route patterns found during continuous observation of the 8 mice using ICG were observed in the 28 mice for the histological experiment (Table 2). Although there were no detours on days 10 and 12, where there was one original route observed in one mouse and no detected routes in 3 mice, the detours appeared from day 15 after the ligation of LVs. No mice developed edema in the abdominal or hindlimb areas during this experiment.

## Microscopic examination

Until day 12, there were no enlarged LVs in the upper subcutaneous tissue and no connection with the collecting LVs between the two ligation sites; however, a few Podoplanin[+] pre-collecting LVs were observed in the upper subcutaneous tissue in the operation side. Serial detour sections were obtained from the specimens on day 30. Enlarged and irregularly shaped detours between the lower portion from the caudal ligation site and the upper portion from the cranial

**Table 1. Number of lymphatic flows in each route pattern after lymphatic ligation until day 30.**

|        | No. of detours | No. of original routes | No. with no detected route |
|--------|----------------|------------------------|----------------------------|
| Day 15 | 2 (25.0)       | 1 (12.5)               | 5 (62.5)                   |
| Day 21 | 4 (50.0)       | 2 (25.0)               | 2 (25.0)                   |
| Day 27 | 5 (62.5)       | 2 (25.0)               | 1 (12.5)                   |
| Day 30 | 5 (62.5)       | 3 (37.5)               | 0 (0.0)                    |

Percentages (%), n = 8 on each day.

**Table 2. Number of lymphatic flow in each route pattern after lymphatic ligation on each observational day.**

|  | No. of detours | No. of original routes | No. of no detected routes |
|---|---|---|---|
| Day 10 | 0 (0.0) | 1 (25.0) | 3 (75.0) |
| Day 12 | 0 (0.0) | 1 (25.0) | 3 (75.0) |
| Day 15 | 2 (50.0) | 0 (0.0) | 2 (50.0) |
| Day 18 | 1 (25.0) | 0 (0.0) | 3 (75.0) |
| Day 21 | 1 (25.0) | 2 (50.0) | 1 (25.0) |
| Day 24 | 2 (50.0) | 1 (25.0) | 1 (25.0) |
| Day 27 | 1 (25.0) | 2 (50.0) | 1 (25.0) |

Percentages (%), n = 4 on each day.

ligation site comprised Podoplanin$^+$ LVs, which were located in subcutaneous upper panniculus carnosus muscle tissue (Fig 3Fi–3Fiv)), and these detours had valves (Fig 3Fiii). The wall of the detours was thin compared with the collecting LV (Fig 3Fx). In the upper portion from the cranial ligation site (upper white arrowhead in Fig 3A), these detours had penetrated the panniculus carnosus muscle and had extended into subcutaneous lower panniculus carnosus muscle tissue and then connected with the collecting LVs that passed through the inguinal LN and ran to the axillary LN (see (a) in Fig 3B and 3Fv–3Fx). Similarly, in the lower portion from the caudal ligation site (lower white arrowhead in Fig 4A), these detours also penetrated the panniculus carnosus muscle and connected with the collecting LVs that passed through the inguinal LN in the subcutaneous tissue under the panniculus carnosus muscle (Fig 4Ei–4Eiii). These detours showed EdU$^-$ lymphatic endothelial cells from days 10 to 30 (Fig 3J–3M), and some EdU$^+$ cells were observed in the epidermis, hair follicles, dermis, and subcutaneous tissues (Fig 3N). Therefore, these detours were developed not through lymphangiogenesis but through an enlargement of the existing pre-collecting LVs. The sham operation showed no enlarged LVs in the upper subcutaneous tissue and no connection was detected with the collecting LVs under the panniculus carnosus muscle. However, although some Podoplanin$^+$ cells were observed in the upper subcutaneous tissue in the sham operation side, vascular structures formed by these cells, which were pre-collecting LVs, had very narrow lumen (Fig 4F and 4G).

The collecting LVs in the lower portion from the cranial ligation site had narrow lumen (Fig 3D and 3E) and the collecting LVs at the caudal ligation site were not detected (Fig 4). These became enlarged in the upper portion from the cranial ligation site (Fig 3Fvi–3Fx) and in the lower portion from the caudal ligation site (Fig 4Eiii).

Capillary LVs were also detected as Podoplanin$^+$ cells in the dermis. They appeared to be enlarged between the two ligation sites (Fig 4H). No connection between the capillary LVs and the pre-collecting LVs was observed in any of the specimens.

In mice with lymphatic detour development, the collecting LV between the two ligations showed considerable obstruction (Fig 3E). In mice with the original route or no detected route, the collecting LV appeared almost normal and unobstructed.

## Discussion

Our study findings showed that, after ligation of a collecting LV, a bridge-like lymphatic detour or a collateral LV formed. The detour comprised the dilatation of an existing pre-collecting LV. The detour was located in the upper subcutaneous tissue and was connected to the upper portion from the cranial ligation and to the lower portion from the caudal ligation. The detour appeared at least 15 days after LV ligation.

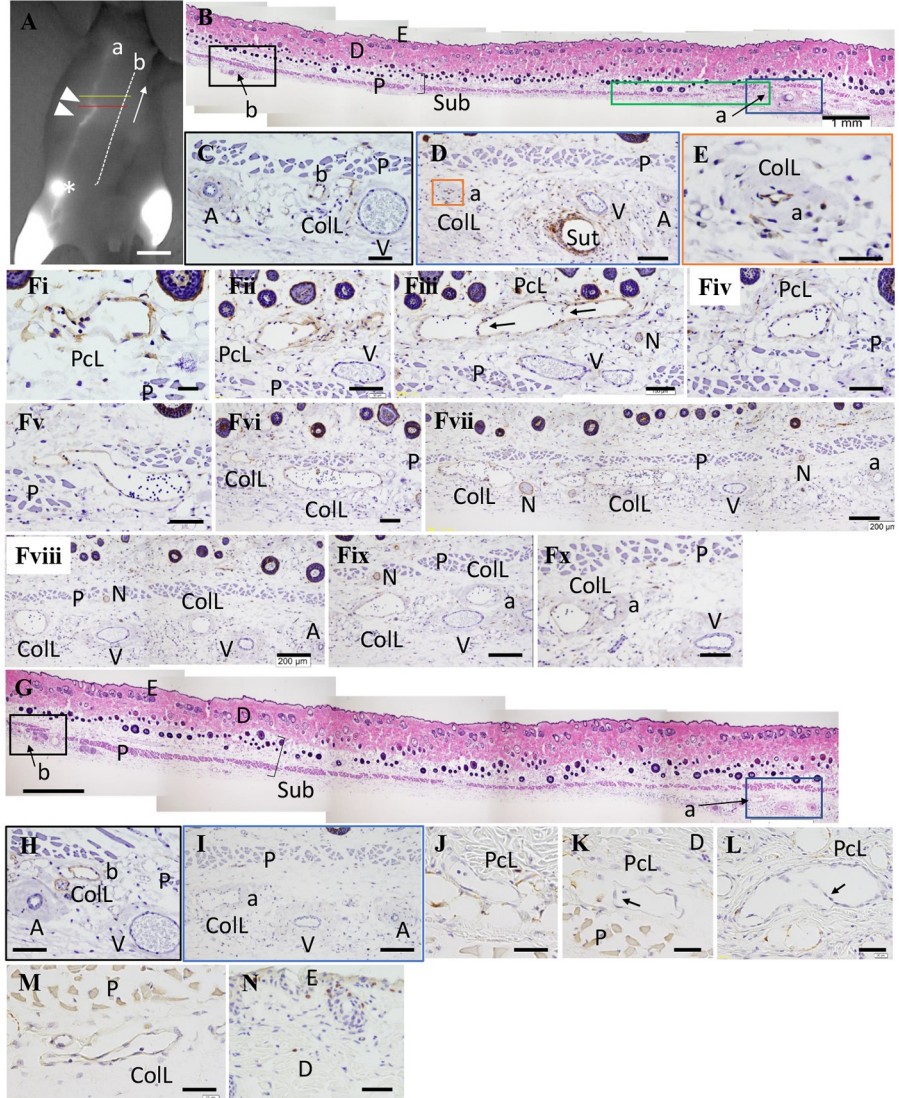

**Fig 3. Representative histological images of detours at the proximal surgical site. (A)** Lymphography of the operation side on day 30. The red line is the starting point for the serial section. The yellow line is the end point for the serial section. The white arrow shows the direction of sectioning; scale bar of 1 cm. **(B), (G)** Hematoxylin and eosin (HE) staining. **(C to F), (H), (I)** Immunostaining with the anti-Podoplanin antibody. **(B)** The section at the red line in Fig 3A; scale bar of 1 mm. **(C)** A high magnification image in the black box in Fig 3B; scale bar of 50 μm. **(D)** A high magnification image in the blue box in Fig 3B. A transverse section of the 6–0 nylon suture (Sut) was observed; scale bar of 100 μm. **(E)** A high magnification image in the orange box in Fig 3D. A small lymphatic canal was observed and smooth muscle forming the wall of the ColL appeared to be thicker; scale bar of 50 μm. **(Fi to Fx)** A serial section of detours in the green box in Fig 3B. It is easy to identify the pre-collecting LV (PcL), stained in brown showing Podoplanin in the lymphatic endothelial cells, that gradually run from above the panniculus carnosus muscle (P) through the muscle to the ColL under the muscle. Many lymphocytes were present in the LVs (round cells stained with purple). Valves can be observed in the PcL (see arrows in Fig Fiii); scale bar of F(i), 20 μm; scale bar of Fii–Fx, 100 μm; scale bar of Fvii–Fix, scale bar of 200 μm. **(G)** The section at the yellow line in Fig 3A; scale bar of 1 mm. **(H)** A high magnification image in the black box in Fig 3G. There was a group of arteries (A), veins (V), and ColL (b) which were adjacent to the LV (a) in (A) shown using ICG; scale bar of 100 μm. **(I)** A high magnification image in the blue box in Fig 3G. Only ColL was observed near the collateral artery (A) and vein (V); scale bar of 200 μm. **(J to N).** Immunostaining to detect EdU. **(J)** A PcL with no EdU stain seen on day 15; scale bar of 40 μm. **(K and L)** PcLs with valves (arrows) with no EdU stain seen on day 30; scale bar of 40 μm. **(M)** A PcL with no EdU stain seen on day 30; scale bar of 40 μm. **(N)** Detection of EdU[+] cells; epidermal cells; hair follicular cells, and; fibroblasts, stained brown in the epidermis and dermis; scale bar of 50 μm. a: LVs passed through the iliac LN and ran into the axillary LN; A: artery; arrows: valve of LV; asterisk: inguinal lymph node (LN); b: LVs that started at the lower abdomen and ran into the axillary LN; closed white arrowheads: ligation sites; ColL: collecting LV; D: dermis; E: epidermis; N: nerve; P: panniculus carnosus muscle; PcL: pre-collecting LV; Sub: subcutaneous tissue; Sut: suture; V: vein.

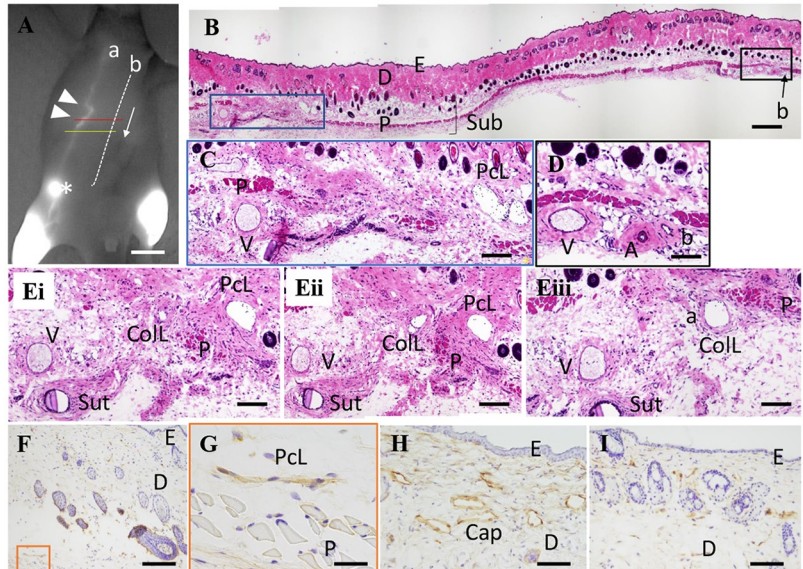

**Fig 4. Representative histological images of detours at the distal surgical site and the sham operation side. (A)**
Lymphography of the operation side on day 30. The red line is the starting point for the serial section. The yellow line
is the end point for the serial section. The white arrow shows the direction of sectioning; scale bar of 1 cm. **(B to E)** HE
staining in the operation side. **(B)** The section at the red line in Fig 4A; scale bar of 200 μm. **(C)** A high magnification
image in the blue box in Fig 4B. Collecting LV (ColL) was not observed and a large pre-collecting LV (PcL) including
lymphocytes in the upper area from the muscle (P) was observed; scale bar of 200 μm. **(D)** A high magnification image
in the black box in Fig 4B; scale bar of 100 μm. **(Ei to–Eiii)** A serial section of detours at the blue box in Fig 4B. The
PcL gradually approaches the ColL and fuses with the ColL (a) **(Eiii)** in the lower area from the muscle (P); scale bar of
100 μm. **(F–G)** Immunostaining with anti-Podoplanin antibody in the sham operation side. A stained thin linear
structure was observed in the subcutaneous tissue beneath the dermis (the orange box); scale bar of F, 200 μm; scale
bar of G, 40 μm. **(G)** A high magnification image in the orange box in Fig 4F. The linear structure is a pre-collecting
LV with no distinguishable lumen; therefore, it is difficult to find the PcL in normal skin stained with HE. **(H)**
Immunostaining with anti-Podoplanin antibody in the operation side; scale bar of 100 μm. Numerous dilated capillary
LVs are seen stained in brown. **(I)** Immunostaining with anti-Podoplanin antibody in the sham operation side. A few
capillary LVs stained brown in the dermis had thinner canals compared with the capillary LVs in H; scale bar of
100 μm. a: the collecting LV that passed through the iliac LN and ran into the axillary LN; A: artery; b: the collecting
LV that started at the lower abdomen and ran into the axillary LN; asterisk: inguinal lymph node (LN); Cap: capillary
LVs; closed white arrowheads: ligation sites; ColL: collecting LV; D: dermis; E: epidermis; P: panniculus carnosus
muscle; PcL: pre-collecting LV; Sub: subcutaneous tissue; Sut: suture; V: vein.

In this study, one lymphatic detour was formed after ligations of the collecting LV at two
points. However, Nakajima et al. [21], Takeno and Fujimoto [15], Yamaji et al. [22], Kwon and
Proce [19], and Ikomi et al. [13] showed that a network-like pattern of LVs and a few collateral
LVs appeared after the LN dissection at least by week 1. Blume et al. [14] reported that a collat-
eral LV was sometimes observed after excision of only the popliteal LN; however, many collat-
eral LVs appeared 1 week after the excision of popliteal LNs within the popliteal fat pad.
Moreover, Ikomi et al. [13] showed that afferent lymphatic recanalization or regeneration was
observed after a 1 mm dissection of the popliteal afferent LVs, but recanalization was not
observed after 3 or 10 mm dissections of the popliteal afferent LVs. Using near-infrared images
and X-ray lymphograms in these reports, there appeared to be no occurrence of pre-collecting
LVs between one end and the other end of the lymphatic dissection, and it was unclear where
the origin and the end of collateral LVs communicated with the remaining LVs. In the present
study, the origin and the end of the lymphatic detours were clear. The lymphatic detour began
from the lower portion from the caudal ligation of the collecting LV and ended at the upper
portion from the cranial ligation of the collecting LV. Therefore, the lymphatic pattern

changed after the ligation of the collecting LVs in this study. The reason for the different findings in previous studies in relation to this study is unclear but may be due to differences in LN dissection that included part of the LVs and blood vessels, and ligation of the LV. More studies are needed to further investigate these different findings.

On days 10 and 12, after the cranial and caudal ligation of the collecting LVs, the original route was observed in one mouse. This may have been due to the loosening of the 6–0 nylon sutures used for the ligation, due to lymph pressure. Bouta et al. [31] reported that lymphatic pumping pressure increased from 11.04 cmH$_2$O under normal conditions to 18.76 cmH$_2$O under expanding LNs, which is similar to what occurs during obstruction of an afferent LV. The re-appearance of the original route took 30 days after ligation, and obstruction of the original route did not reoccur. The lymphatic detours we observed remained throughout the course of the study period. However, it was earlier than 15 days and later than 30 days before the lymphatic detours appeared after the ligation, while collateral LVs appeared from 1 to 4 weeks after LN dissection [13, 15, 19, 21, 22], indicating that our findings were consistent with those of previous studies.

Pre-collecting LVs (pre-collectors) have valves and are located between capillary LVs and collecting LVs. They collect lymph from the capillary network and direct it into the collecting LVs (collectors) [32]. Pre-collecting LVs are characterized with alternating sections of absorbing structures and sections with a developed muscular layer. This suggests that they contribute to fluid absorption, capillary lymphatics, and to lymph propulsion, in a manner similar to collecting LVs [33]. In our study, detours were observed in the upper subcutaneous tissue after lymphatic ligation, and most detours appeared between two collecting LVs in the abdomen. These findings were consistent with those in our previous study [22]. Detours detected in the present study had Podoplanin$^+$ cells and valves, which were observed between capillary LVs in the dermis and collecting LVs in the lower subcutaneous tissue, and were connected to the collecting LVs. They had expanded lumen and they drained lymph at the operative field, since the collecting LVs at the two ligation sites either had narrow lumen or the lumen were not detected; therefore, their ability to transport lymph may have been reduced due to lymphatic ligation. These findings support those of a previous study that examined the function of pre-collecting LVs [33]. Consequently, we found that these detours were pre-collecting LVs that functioned as collecting LVs after lymphatic obstruction. It is difficult to detect pre-collecting LVs in the upper subcutaneous tissue in skin with normal lymphatic transport because pre-collecting LVs normally exhibit no enlargement of the lumen. Lymphatic ligation facilitates detection of pre-collecting LVs.

Lymphangiogenesis was evaluated through the expression of EdU. An EdU injection was performed twice, over two days, because the doubling time of human lymphatic endothelial cells is between 15 and 48 h (no data is available concerning the doubling time of lymphatic endothelial cells in mice) [34]. Some cells, such as the keratinocytes and hair follicles, incorporated EdU; however, EdU$^+$ cells were not observed in the detours. Moreover, Tammela et al. [27] reported that, after the application of VEGF-C to the surgical lymph node excision site and removal of associated collecting LVs, mature collecting LVs appeared within 2 months. These observations suggest that detours after lymphatic ligation were not formed through lymphangiogenesis. A possible formation process concerning lymphatic detours could occur as follows:

1. Lymph drained from the hindlimb reaches the caudal ligation site or the obstruction site of the collecting LV.

2. Lymph flows back to the pre-collecting LV near the lower site from the caudal ligation.

3. Lymph back flow compresses the valve in the pre-collecting LV after lymphatic ligation, suddenly impairing the valve, and the lymph is drained into the pre-collecting LV, which is finally drained into the pre-collecting LV communicating to the upper site from the cranial ligation of the colleting LV.

4. Bridge-like lymphatic detours are then formed. Additionally, the dilated capillary LVs appearing in the dermis and covering the cranial and caudal ligation of the collecting LVs seem to be partly emerge due to lymph back flow from the pre-collecting LVs.

Abe [10] observed that, in patients with mild or no lymphedema after LN resection, lymph flowed from the original route to the supraclavicular LNs, or from a collateral LV to the shoulder or lateral chest wall. Maegawa et al. [11] identified a link between obvious inguinal LNs and lymphatics. Mikami et al. [5] reported the appearance of LNs around the clavicle and collateral LVs. Szuba et al. [7] showed that LNs were present in differing locations. These studies did not describe how new LV appeared across the area of LN dissections; therefore, this may imply that the regeneration of collecting LVs did not occur or that this was difficult to determine in the LN dissection area, but that formation of collateral LVs, as in the lymphatic detours observed in this study around the area of lymphadenectomy, are possible through changing the LV pathways. Suami et al. [25] showed that lymphatic pathway changes could occur, that is, lymph pathways had connected the superficial and deep lymphatic systems in a human cadaver after blockage of the collecting LVs, and this connection had not previously been observed in a healthy arm. Moreover, Suami et al. [25] proposed different types of lymphatic pathway changes following blockage in lymph collecting vessels, which could be very useful for forming collateral LVs or lymphatic detours. As lymphatic pathway changes were also observed in our study using mice, our model might be useful to clarify the mechanism of reconstruction of the lymphatic route and to study the prevention and treatment of lymphedema. Further studies are warranted to explore reconstruction mechanisms concerning lymphatic routes and to determine how lymphedema develops or does not develop in humans.

This study had some limitations. We did not evaluate the valve function of the pre-collecting LVs. Lymph flow in mice that displayed the original route was not assessed and it was not clear how the lymph was drained via this route. Furthermore, we did not perform clinical correlation or functioning tests concerning the lymphatic detour.

## Conclusion

Our study findings showed that, after cranial and caudal ligations of a collecting LV, the bridge-like lymphatic detour connecting the lower portion from the caudal ligation and the upper portion from the cranial ligation was the dilated pre-collecting LV. As this detour usefully functioned to drain lymph after LV obstruction, this finding suggests that, should a bridge-like detour develop in a patient with lymphedema, it may be possible to cure lymphedema. Therefore, future studies investigating the developmental mechanisms of lymphatic detour for treatment and for effective care in relation to lymphedema in humans are needed.

## Acknowledgments

We would like to thank Editage (www.editage.com) for English language editing.

## Author Contributions

**Conceptualization:** Kimi Asano, Yukari Nakajima, Toshio Nakatani.

**Data curation:** Kimi Asano, Yukari Nakajima.

**Formal analysis:** Yukari Nakajima.

**Funding acquisition:** Kimi Asano, Yukari Nakajima.

**Investigation:** Kimi Asano, Yukari Nakajima.

**Methodology:** Kimi Asano, Yukari Nakajima, Kanae Mukai, Toshio Nakatani.

**Supervision:** Kanae Mukai, Tamae Urai, Mayumi Okuwa, Junko Sugama, Chizuko Konya, Toshio Nakatani.

**Visualization:** Kimi Asano, Yukari Nakajima.

**Writing – original draft:** Kimi Asano, Yukari Nakajima.

**Writing – review & editing:** Kanae Mukai, Tamae Urai, Mayumi Okuwa, Junko Sugama, Chizuko Konya, Toshio Nakatani.

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
