## [Decision Letter · Decision Letter 0]

16 Sep 2019

PONE-D-19-17179

Connecting lymphatic vessels form detours upon obstruction of lymphatic flow and function as collecting lymphatic vessels

PLOS ONE

Dear Dr. Nakatani,

Thank you for submitting your manuscript to PLOS ONE. After careful consideration, we feel that it has merit but does not fully meet PLOS ONE’s publication criteria as it currently stands. Therefore, we invite you to submit a revised version of the manuscript that addresses the points raised during the review process.

As you will recognize from the comments of the reviewers they raised points of critique, especially regarding design of the study and statistical issues as well as rational of the study.

We would appreciate receiving your revised manuscript by within 2 months. To enhance the reproducibility of your results, we recommend that if applicable you deposit your laboratory protocols in protocols.io, where a protocol can be assigned its own identifier (DOI) such that it can be cited independently in the future. For instructions see: http://journals.plos.org/plosone/s/submission-guidelines#loc-laboratory-protocols

We look forward to receiving your revised manuscript.

Kind regards,

Rudolf Kirchmair

Academic Editor

PLOS ONE

Journal Requirements:

2. In your Methods section, please describe any postoperative care or analgesia provided to the animals used in your study.

Additional Editor Comments (if provided):

Reviewers' comments:

Reviewer's Responses to Questions

**Comments to the Author**

1. Is the manuscript technically sound, and do the data support the conclusions?

Reviewer #1: Partly

Reviewer #2: Partly

2. Has the statistical analysis been performed appropriately and rigorously? 

Reviewer #1: No

Reviewer #2: N/A

3. Have the authors made all data underlying the findings in their manuscript fully available?

Reviewer #1: Yes

Reviewer #2: Yes

4. Is the manuscript presented in an intelligible fashion and written in standard English?

Reviewer #1: Yes

Reviewer #2: Yes

5. Review Comments to the Author

Reviewer #1: This is a very interesting manuscript about the formation of detours by connective lymphatic vessels after obstruction of lymphatic flow in mice. In addition, it is a current and important problem for patients suffering from lymphedema. The Abstract and Introduction are in line with the work presented and are clearly organized. However, the authors should clarify in the Material and Methods section if the statistical analyzes were performed and how they were made. In addition, the authors do not mention these analyzes in the Results section. Therefore, the statistical analyzes details should be clarified to ensure that readers understand exactly what the researchers studied. Also, the authors should include more information that clarifies and justifies their choice of methods. In addition, the authors need to be bolder and more analytical in the Discussão section, using more area references to expand the discussion and make it even more interesting. Regarding the Conclusions section, conclusions must be drawn appropriately based on the data presented, not just a repetition of the data found.

Minor points:

1. Line 93, page 6. It is written "the cranial ligation site and the caudal ligation site." Were these points chosen in a specific or random way? It is not clear.

2. Line 127, page 8. "... with 1.75% isoflurane..." is written in different letters from the rest of the text. Please, check the text formatting.

3. Line 148, page 9. It is written that "... stapled onto transparent plastic sheets...". Did the plastic allow the correct fixation of the collected material?

4. Line 287, page 17. Reference 15 is not enclosed in square brackets as the rest of the references.

5. Figure legends need to be better organized. For example, after (A) do not use the end point [(A) Evans blue dye ...]. Another example, do not use a semicolon (;) to separate the scale bar from its size [you can use Scale bar of 1 cm]. To indicate objects in the figure, do not separate their name from their function using a semicolon (;), you can use a colon (:) and to separate the different objects in the figure you use semicolons (;) [example: Open white arrowheads: incision site; asterisk: inguinal LN]. Please you should make these changes in all captions of the figures.

Reviewer #2: The authors observed detour of lymphatics (collateral formation?) in mice after lymphadenectomy in their previous study (ref 8). They aimed to investigate this detour phenomena in this current study by ligation of the lymphatic vessel in distal and proximal portions in 51 mice. The authors used ICG lymphography to image the detour formation and performed histological analysis of the detoured vessel. 5-ethynyl-2’-deoxyuridine (EdU) was used to evaluate any evidence of lymphangiogenesis. The authors found that detour was formed by 30 days after ligation and the detoured vessels were not formed by lymphangiogenesis. They concluded that the detour of lymphatic vessels might be useful for effective care of lymphedema.

Although investigation of structural changes of the lymphatic vessel after segmented obstruction with ligation in mice may be unique, there are several fundamental questions about this study.

In line 91, the authors described that they developed detours by excision of the inguinal lymph node but the pattern of detours by this surgery was unexpected in their previous study. However, they did not explain what was unexpected. This finding was the reason for current study design and therefore it requires further explanation.

There are other concerns required further explanation.

From line 66 to 72, patients with lymphedema demonstrated collateral formation and structural changes but the authors said detours of the lymphatics after the lymphadenectomy in mice suggested that these detours prevented lymphedema. There was contradiction between stories in patients and mice.

From line 82 to 87, 8 mice were used for lymphography and histological evaluation on day 30. 28 mice were divided into 7 groups (n-4 each) and used by the same evaluation at different time points (Day 10, 12, 15, 18, 21, 24, and 27). However, Results in Table 1 showed 8 mice studied at different time points (Day 15, 12, 27 and 30). Procedures in 8 mice were missing in the Materials and Methods section.

In line 10, lymphatic routes after the ligation were divided into three patterns; detours, original route and no detection. The detours were discussed in the Discussion but the other two patterns were not discussed. As for technical aspect, I think 6-0 nylon was too thick to ligate the lymphatic vessels in mice. I recommend using 9-0 or 10-0 for this purpose.

In line 232, the authors concluded that these detours were not developed by lymphangiogenesis. This might be true for this study, however I don’t think EdU is adequate enough to appreciate lymphangiogenesis at this stage. There is a substantial body of research that highlights lymphangiogenesis after surgical procedures in different animals including mice, rats, rabbits, digs, pigs and sheep. It is insufficient to conclude the absence of lymphangiogenesis on the basis of negative EdU cells alone.

In line 313, the authors described no obvious evidence of lymphatic reconstruction of the lymphatic route in humans. This statement is inaccurate. Alterations of the lymphatic pathways in patients with lymphedema have been examined with lymphangiography, lymphoscintigraphy, ICG lymphography, MR lymphography, and SPECT/CT. Each clinical imaging has specific criteria to diagnose lymphedema.

Overall, the manuscript requires further explanation about background of the study design in Introduction and detailed procedures in Materials and Methods. A strong point of this study was combination of ICG lymphography and immune-histochemical analysis and weak points were clinical correlations and lack of functional testing.

6. PLOS authors have the option to publish the peer review history of their article (what does this mean?). If published, this will include your full peer review and any attached files.

Reviewer #1: No

Reviewer #2: No

---

## [Author Response · Author response to Decision Letter 0]

6 Nov 2019

Dear Editor:

Thank you and the reviewers for these important comments and advice concerning this manuscript. We have revised our paper according to the reviewers’ comments. 

We have revised the following sections:

Discussion: We have described the formation and characteristics of the lymphatic detour from the present results, we have added references, and we discuss our findings that show the detour is useful to decrease lymphedema.

Introduction: As the discussion has been revised, we have also revised the Introduction section accordingly and have add animal and human references.

We have got English proofreading in our revised manuscript, and revised the following sections:

Title of this manuscript: We have changed “Connecting lymphatic vessels form detours upon obstruction of lymphatic flow and function as collecting lymphatic vessels.” to “Connecting lymphatic vessels form detours following obstruction of lymphatic flow and function as collecting lymphatic vessels.” 

Acknowledgements: We have added “We would like to thank Editage (www.editage.com) for English language editing.” on page 27, lines 471-472 because the company requests us.

We kindly request you to please review our newly revised manuscript. 

Reviewer #1:

 This is a very interesting manuscript about the formation of detours by connective lymphatic vessels after obstruction of lymphatic flow in mice. In addition, it is a current and important problem for patients suffering from lymphedema. The Abstract and Introduction are in line with the work presented and are clearly organized. 

Q1. However, the authors should clarify in the Material and Methods section if the statistical analyzes were performed and how they were made. In addition, the authors do not mention these analyzes in the Results section. Therefore, the statistical analyzes details should be clarified to ensure that readers understand exactly what the researchers studied. Also, the authors should include more information that clarifies and justifies their choice of methods. 

Response 1.

Thank you for your comments.

As you have mentioned, it is important to state how statistical analyses were performed in the Materials and Methods section. However, as Reviewer 2 has also judged, we consider that statistical analysis was not applicable in this study. We provide the following two reasons for not having undertaken statistical analyses in this study.

First, we do not compare any results from Tables 1 and 2 in the Results and Discussion sections. We only classified the patterns of lymphatic flow after lymphatic ligation, and we have focused mainly on detours after lymphatic ligation in the Results and Discussion sections.

Second, we were not able to perform statistical analysis concerning results shown in Tables 1 and 2 because the numbers of each pattern are too small (e.g. 0 or 1).

As such, we did not perform any statistical analysis in this study.

Additionally, we have revised the ICG lymphography results. We consider that the description is now easier to understand. We refer you to pages 13-14, lines 226-248.

Q2. In addition, the authors need to be bolder and more analytical in the Discussão section, using more area references to expand the discussion and make it even more interesting. Regarding the Conclusions section, conclusions must be drawn appropriately based on the data presented, not just a repetition of the data found.

Response 2. 

We have revised the Introduction and Discussion sections in accordance with the reviewers’ comments, which we consider have now greatly improved our manuscript. Kindly refer to our newly revised changes.

Minor points:

1. Line 93, page 6. It is written "the cranial ligation site and the caudal ligation site." Were these points chosen in a specific or random way? It is not clear.

Response 3.

We have added a sentence on page 9, lines 141-142, and our changes are highlighted in red.

2. Line 127, page 8. "... with 1.75% isoflurane..." is written in different letters from the rest of the text. Please, check the text formatting.

Response 4.

We have now corrected this on page 11, line 180, and our changes are highlighted in red.

3. Line 148, page 9. It is written that "... stapled onto transparent plastic sheets...". Did the plastic allow the correct fixation of the collected material?

Response 5.

We have added a sentence on page 12, lines 201- 204, and our changes are highlighted in red.

4. Line 287, page 17. Reference 15 is not enclosed in square brackets as the rest of the references.

Response 6.

Number of Refence 15 was changed to Reference 32. We have enclosed reference 32 in square brackets on page 23, line 399, and our changes are highlighted in red.

5. Figure legends need to be better organized. For example, after (A) do not use the end point [(A) Evans blue dye ...]. Another example, do not use a semicolon (;) to separate the scale bar from its size [you can use Scale bar of 1 cm]. To indicate objects in the figure, do not separate their name from their function using a semicolon (;), you can use a colon (:) and to separate the different objects in the figure you use semicolons (;) [example: Open white arrowheads: incision site; asterisk: inguinal LN]. Please you should make these changes in all captions of the figures.

Response 7.

We have undertaken to make all of the recommended amendments to the figure legends, and our changes are highlighted in red.

Reviewer #2:

 The authors observed detour of lymphatics (collateral formation?) in mice after lymphadenectomy in their previous study (ref 8). They aimed to investigate this detour phenomena in this current study by ligation of the lymphatic vessel in distal and proximal portions in 51 mice. The authors used ICG lymphography to image the detour formation and performed histological analysis of the detoured vessel. 5-ethynyl-2’-deoxyuridine (EdU) was used to evaluate any evidence of lymphangiogenesis. The authors found that detour was formed by 30 days after ligation and the detoured vessels were not formed by lymphangiogenesis. They concluded that the detour of lymphatic vessels might be useful for effective care of lymphedema.

Although investigation of structural changes of the lymphatic vessel after segmented obstruction with ligation in mice may be unique, there are several fundamental questions about this study.

Q1. In line 91, the authors described that they developed detours by excision of the inguinal lymph node but the pattern of detours by this surgery was unexpected in their previous study. However, they did not explain what was unexpected. This finding was the reason for current study design and therefore it requires further explanation.

Response 1. 

Thank you for your comments.

We have revised the sentence on pages 8-9, lines 135-139, and our changes are highlighted in red. 

Q2. From line 66 to 72, patients with lymphedema demonstrated collateral formation and structural changes but the authors said detours of the lymphatics after the lymphadenectomy in mice suggested that these detours prevented lymphedema. There was contradiction between stories in patients and mice.

Response 2.

We have made significant changes to the Introduction, which is now ready for your further consideration. Pages 4-6, lines 53-82 correspond to almost all of your comments.

Q3. From line 82 to 87, 8 mice were used for lymphography and histological evaluation on day 30. 28 mice were divided into 7 groups (n-4 each) and used by the same evaluation at different time points (Day 10, 12, 15, 18, 21, 24, and 27). However, Results in Table 1 showed 8 mice studied at different time points (Day 15, 12, 27 and 30). Procedures in 8 mice were missing in the Materials and Methods section.

Response 3. 

We have revised the animals in the Materials and Methods section, on page 8, lines 121-132, and our changes are highlighted in red. Kindly refer to our newly revised changes. 

Q4. In line 10, lymphatic routes after the ligation were divided into three patterns; detours, original route and no detection. The detours were discussed in the Discussion but the other two patterns were not discussed. As for technical aspect, I think 6-0 nylon was too thick to ligate the lymphatic vessels in mice. I recommend using 9-0 or 10-0 for this purpose.

Response 4.

We have made many revisions to the Discussion section. Kindly refer to pages 21-23, lines 367-396, in response to your comments. In our next experiment, we intend to use a 9-0 or 10-0 suture.

Q5. In line 232, the authors concluded that these detours were not developed by lymphangiogenesis. This might be true for this study, however I don’t think EdU is adequate enough to appreciate lymphangiogenesis at this stage. There is a substantial body of research that highlights lymphangiogenesis after surgical procedures in different animals including mice, rats, rabbits, digs, pigs and sheep. It is insufficient to conclude the absence of lymphangiogenesis on the basis of negative EdU cells alone.

Response 5.

Thank you for your valuable advice. We greatly appreciate your comments, which we agree with. In future studies, we intend to use another method to detect lymphangiogenesis or dilatation of existing connecting lymphatics forming lymphatic detours. 

Q6. In line 313, the authors described no obvious evidence of lymphatic reconstruction of the lymphatic route in humans. This statement is inaccurate. Alterations of the lymphatic pathways in patients with lymphedema have been examined with lymphangiography, lymphoscintigraphy, ICG lymphography, MR lymphography, and SPECT/CT. Each clinical imaging has specific criteria to diagnose lymphedema.

Response 6.

In response to your helpful comments, we have significantly revised the Discussion and Introduction sections. We have added a lymphatic image of human lymphedema to pages 4-5, lines 53-74 of the Introduction section, and we have revised part of the Discussion section on pages 25-26, lines 436-454 for your consideration. 

Q8. Overall, the manuscript requires further explanation about background of the study design in Introduction and detailed procedures in Materials and Methods. A strong point of this study was combination of ICG lymphography and immune-histochemical analysis and weak points were clinical correlations and lack of functional testing.

Response 8.

We have considerably revised the Introduction and Discussion sections according to the reviewers’ comments. We consider that the reviewers’ comments have helped us to improve our manuscript.

---

## [Decision Letter · Decision Letter 1]

27 Nov 2019

PONE-D-19-17179R1

Connecting lymphatic vessels form detours following obstruction of lymphatic flow and function as collecting lymphatic vessels

PLOS ONE

Dear Dr. Nakatani,

Thank you for submitting your manuscript to PLOS ONE. After careful consideration, we feel that it has merit but does not fully meet PLOS ONE’s publication criteria as it currently stands. Therefore, we invite you to submit a revised version of the manuscript that addresses the points raised during the review process.

As you will see from comments of Reviewer 2 still Major Points of critique were raised especially regarding presentation of the manuscript.

We would appreciate receiving your revised manuscript within two months. To enhance the reproducibility of your results, we recommend that if applicable you deposit your laboratory protocols in protocols.io, where a protocol can be assigned its own identifier (DOI) such that it can be cited independently in the future. For instructions see: http://journals.plos.org/plosone/s/submission-guidelines#loc-laboratory-protocols

We look forward to receiving your revised manuscript.

Kind regards,

Rudolf Kirchmair

Academic Editor

PLOS ONE

Reviewers' comments:

Reviewer's Responses to Questions

**Comments to the Author**

1. If the authors have adequately addressed your comments raised in a previous round of review and you feel that this manuscript is now acceptable for publication, you may indicate that here to bypass the “Comments to the Author” section, enter your conflict of interest statement in the “Confidential to Editor” section, and submit your "Accept" recommendation.

Reviewer #1: All comments have been addressed

Reviewer #2: (No Response)

2. Is the manuscript technically sound, and do the data support the conclusions?

Reviewer #1: Yes

Reviewer #2: Yes

3. Has the statistical analysis been performed appropriately and rigorously? 

Reviewer #1: N/A

Reviewer #2: N/A

4. Have the authors made all data underlying the findings in their manuscript fully available?

Reviewer #1: Yes

Reviewer #2: Yes

5. Is the manuscript presented in an intelligible fashion and written in standard English?

Reviewer #1: Yes

Reviewer #2: No

6. Review Comments to the Author

Reviewer #1: The "Introduction" section should be formatted to "justified" as it is formatted "on the left". At line 175, in the caption of Figure 1, the semicolon after "asterisk" should be changed to two points. In line 258, in the caption of Figure 2, a period should be placed at the end of the sentence instead of a semicolon. In lines 318 and 319, in the caption of Figure 3, in the phrase "scale bar of F (vii)–F (ix); scale bar of 200 μm", the semicolon must be replaced by a comma. In line 332, in the caption of Figure 3, a space must be given between "LV; D" (LV;D).

Reviewer #2: The introduction from page 4 to page 7 in the new manuscript was significantly changed from the initial draft that was 1.5 page long. However, the lengthy introduction diluted the primary aim of this animal experiment and failed to focus on the importance of this study. Some paragraphs seem to be more suitable in the discussion section.

Two sentences in the background of the abstract remained similar to the previous draft and therefore it lost connection to the revised introduction.

Reviewer comments by reviewer 1 and 2 did not suggest such significant changes throughout the whole paper. I recommend that the authors should consider staying point-to-point with their response to address reviewers’ suggestions or submitting it as a new submission.

Several new sentences are unclear. In page 5, line 77-79, “It is unclear whether a lymphatic detour…” This sentence may describe important background information about the current study but I don’t understand the difference between “lymphoangiogenesis” and “changing lymphatic patterns where…”. In page 7, lines 106-108, “In this study, we aimed to determine…” This sentence seems a key message to stress the aim of this study. However, it doesn’t describe logical meanings and has grammatical errors.

The authors should clarify the terminology to describe the lymphatic structures in the introduction section. In page 23, line 397, they described that “connecting LVs” are as same as “pre-collectors”. I am uncertain that connecting is the right anatomical word to describe pre-collector. Even if the word is correct, they should avoid using it for this study. “Connecting” has another meaning, “connecting between the lymphatic gap” and therefore the readers may confuse the meaning with “detour”. To avoid confusion, I recommend pre-collector is a more suitable word for this manuscript. For example, in page 21, line 363-364, “The detour comprised…” The authors wrote “connecting or pre-collecting LV” but this should be “pre-collecting (connecting) LV”. Another example in page 22, line 376, “there appeared to be no occurrence of connecting LVs…” The author used the word “connecting” as meaning of filling the gap. It appears that the authors were confused as well.

Introduction section must be revised but the rest of the manuscript seems to be improved. This study seems to provide some new meaningful findings regarding how the lymphatic system is reconstructed after surgical intervention. However, the inaccurate English writing prevents understanding of the true scientific value of this study.

I suggest that the manuscript should be edited by English speaking persons to retain the scientific accuracy of the study before the review process.

7. PLOS authors have the option to publish the peer review history of their article (what does this mean?). If published, this will include your full peer review and any attached files.

Reviewer #1: No

Reviewer #2: Yes: Hiroo Suami

---

## [Author Response · Author response to Decision Letter 1]

9 Dec 2019

Dear Editor:

We thank the editor and the reviewers for their important comments and advice concerning our manuscript. We have revised our paper, especially the introduction, according to the reviewers’ and editor’s comments. 

We hereby submit the revised manuscript both with track changes and unmarked. We kindly request you to please review our newly revised manuscript.

Our point-by-point responses are indicated below.

Reviewer 1

Q1. The "Introduction" section should be formatted to "justified" as it is formatted "on the left".

R1. We thank the reviewer for the astute observation, we have now formatted as justified.

Q2. At line 175, in the caption of Figure 1, the semicolon after "asterisk" should be changed to two points.

R2. We thank the reviewer for the astute observation, we changed the semicolon to colon, in line 157 in the revised manuscript.

Q3. In line 258, in the caption of Figure 2, a period should be placed at the end of the sentence instead of a semicolon.

R3. We thank the reviewer for the astute observation, we changed the semicolon to period, in line 241 in the revised manuscript.

Q4. In lines 318 and 319, in the caption of Figure 3, in the phrase "scale bar of F (vii)–F (ix); scale bar of 200 μm", the semicolon must be replaced by a comma.

R4. We thank the reviewer for the astute observation, we changed the semicolon to comma, in line at line 302 in the revised manuscript.

Q5. In line 332, in the caption of Figure 3, a space must be given between "LV; D" (LV;D).

R5. We thank the reviewer for the astute observation, we changed LV;D to LV; D in line 314 in the revised manuscript.

Reviewer 2

Q1. The introduction from page 4 to page 7 in the new manuscript was significantly changed from the initial draft that was 1.5 page long. However, the lengthy introduction diluted the primary aim of this animal experiment and failed to focus on the importance of this study. Some paragraphs seem to be more suitable in the discussion section.

R1. We thank the reviewer for this important observation, we revised the introduction and shortened it from four pages to three pages. The deleted sentences as well as the revised sentences are shown with track changes in the revised manuscript.

Q2. Two sentences in the background of the abstract remained similar to the previous draft and therefore it lost connection to the revised introduction. 

R2. As we reduced the introduction to clarify the aim of this research, we believe that the two sentences in the abstract are now well-aligned with the introduction. As we described in the first paragraph, the collateral lymphatic vessels or lymphatic detours are important for the decrease in lymphedema; we included this in our previous paper [21]. Furthermore, along with the importance of the process of forming lymphatic detours included in the second paragraph of the introduction show the connection with the two sentences in the background section of the abstract. 

Editor

Q1. Reviewer comments by reviewer 1 and 2 did not suggest such significant changes throughout the whole paper. I recommend that the authors should consider staying point-to-point with their response to address reviewers’ suggestions or submitting it as a new submission.

R1. We thank the editor for these important comments. As we mentioned above, we have revised our manuscript based on the reviewers’ comments, we included the point-by-point responses, and also submitted the revised manuscript again.

Q2. In page 5, line 77-79, “It is unclear whether a lymphatic detour…” This sentence may describe important background information about the current study but I don’t understand the difference between “lymphoangiogenesis” and “changing lymphatic patterns where…”.

R2. We thank the editor for these important comments. As we have now revised the introduction, we have clarified further in pages 5-6, lines 69 to 85, third paragraph in the introduction section of the revised manuscript. We hope that the difference between lymphoangiogenesis and changing lymphatic patterns can now be better understood.

Q3. In page 7, lines 106-108, “In this study, we aimed to determine…” This sentence seems a key message to stress the aim of this study. However, it doesn’t describe logical meanings and has grammatical errors.

R3. We have revised these sentences in page 6, from lines 86 to 91 in the revised manuscript. Based on our revised sentences in the introduction, we hope that the aim is now acceptable. In the third paragraph of the introduction we described the appearance, position in the skin, and formation of lymphatic detours logically or step by step.

Q4. The authors should clarify the terminology to describe the lymphatic structures in the introduction section. In page 23, line 397, they described that “connecting LVs” are as same as “pre-collectors”. I am uncertain that connecting is the right anatomical word to describe pre-collector. Even if the word is correct, they should avoid using it for this study. “Connecting” has another meaning, “connecting between the lymphatic gap” and therefore the readers may confuse the meaning with “detour”. To avoid confusion, I recommend pre-collector is a more suitable word for this manuscript. For example, in page 21, line 363-364, “The detour comprised…” The authors wrote “connecting or pre-collecting LV” but this should be “pre-collecting (connecting) LV”. Another example in page 22, line 376, “there appeared to be no occurrence of connecting LVs…” The author used the word “connecting” as meaning of filling the gap. It appears that the authors were confused as well.

R4. We changed connecting LVs to pre-collecting LV according to your suggestion.

Q5. I suggest that the manuscript should be edited by English speaking persons to retain the scientific accuracy of the study before the review process.

R5. We thank the editor for these relevant comments. As you suggested, a native English language speaker from an English editing company has edited our revised manuscript and has improved grammar and scientific accuracy.

---

## [Decision Letter · Decision Letter 2]

31 Dec 2019

Pre-collecting lymphatic vessels form detours following obstruction of lymphatic flow and function as collecting lymphatic vessels

PONE-D-19-17179R2

Dear Dr. Nakatani,

We are pleased to inform you that your manuscript has been judged scientifically suitable for publication and will be formally accepted for publication once it complies with all outstanding technical requirements.

With kind regards,

Rudolf Kirchmair

Academic Editor

PLOS ONE

Additional Editor Comments (optional):

Reviewers' comments:

Reviewer's Responses to Questions

**Comments to the Author**

1. If the authors have adequately addressed your comments raised in a previous round of review and you feel that this manuscript is now acceptable for publication, you may indicate that here to bypass the “Comments to the Author” section, enter your conflict of interest statement in the “Confidential to Editor” section, and submit your "Accept" recommendation.

Reviewer #1: All comments have been addressed

Reviewer #2: All comments have been addressed

2. Is the manuscript technically sound, and do the data support the conclusions?

Reviewer #1: Yes

Reviewer #2: Yes

3. Has the statistical analysis been performed appropriately and rigorously? 

Reviewer #1: N/A

Reviewer #2: N/A

4. Have the authors made all data underlying the findings in their manuscript fully available?

Reviewer #1: Yes

Reviewer #2: Yes

5. Is the manuscript presented in an intelligible fashion and written in standard English?

Reviewer #1: Yes

Reviewer #2: Yes

6. Review Comments to the Author

Reviewer #1: (No Response)

Reviewer #2: The authors addressed all reviewers’ suggestions. The manuscript became easier to read and improved in English writing.

7. PLOS authors have the option to publish the peer review history of their article (what does this mean?). If published, this will include your full peer review and any attached files.

Reviewer #1: No

Reviewer #2: No

---

## [Editor Report · Acceptance letter]

6 Jan 2020

PONE-D-19-17179R2 

Pre-collecting lymphatic vessels form detours following obstruction of lymphatic flow and function as collecting lymphatic vessels 

Dear Dr. Nakatani:

I am pleased to inform you that your manuscript has been deemed suitable for publication in PLOS ONE. Congratulations! Your manuscript is now with our production department. 

With kind regards,

on behalf of

Prof Rudolf Kirchmair 

Academic Editor

PLOS ONE